# Best-Arm Identification in Linear Bandits

**Marta Soare**       **Alessandro Lazaric**       **Rémi Munos**[*][†]
INRIA Lille – Nord Europe, SequeL Team
{marta.soare,alessandro.lazaric,remi.munos}@inria.fr

## Abstract

We study the best-arm identification problem in linear bandit, where the rewards of the arms depend linearly on an unknown parameter $\theta^*$ and the objective is to return the arm with the largest reward. We characterize the complexity of the problem and introduce sample allocation strategies that pull arms to identify the best arm with a fixed confidence, while minimizing the sample budget. In particular, we show the importance of exploiting the global linear structure to improve the estimate of the reward of near-optimal arms. We analyze the proposed strategies and compare their empirical performance. Finally, as a by-product of our analysis, we point out the connection to the $G$-optimality criterion used in optimal experimental design.

## 1   Introduction

The stochastic multi-armed bandit problem (MAB) [16] offers a simple formalization for the study of sequential design of experiments. In the standard model, a learner sequentially chooses an arm out of $K$ and receives a reward drawn from a fixed, unknown distribution relative to the chosen arm. While most of the literature in bandit theory focused on the problem of maximization of cumulative rewards, where the learner needs to trade-off exploration and exploitation, recently the *pure exploration* setting [5] has gained a lot of attention. Here, the learner uses the available budget to identify as accurately as possible the best arm, without trying to maximize the sum of rewards. Although many results are by now available in a wide range of settings (e.g., best-arm identification with fixed budget [2, 11] and fixed confidence [7], subset selection [6, 12], and multi-bandit [9]), most of the work considered only the multi-armed setting, with $K$ independent arms.

An interesting variant of the MAB setup is the stochastic *linear bandit* problem (LB), introduced in [3]. In the LB setting, the input space $\mathcal{X}$ is a subset of $\mathbb{R}^d$ and when pulling an arm $x$, the learner observes a reward whose expected value is a linear combination of $x$ and an unknown parameter $\theta^* \in \mathbb{R}^d$. Due to the linear structure of the problem, pulling an arm gives information about the parameter $\theta^*$ and indirectly, about the value of other arms. Therefore, the estimation of $K$ mean-rewards is replaced by the estimation of the $d$ features of $\theta^*$. While in the exploration-exploitation setting the LB has been widely studied both in theory and in practice (e.g., [1, 14]), in this paper we focus on the pure-exploration scenario.

The fundamental difference between the MAB and the LB best-arm identification strategies stems from the fact that in MAB an arm is no longer pulled as soon as its sub-optimality is evident (in high probability), while in the LB setting even a sub-optimal arm may offer valuable information about the parameter vector $\theta^*$ and thus improve the accuracy of the estimation in discriminating among near-optimal arms. For instance, consider the situation when $K-2$ out of $K$ arms are already discarded. In order to identify the best arm, MAB algorithms would concentrate the sampling on the two remaining arms to increase the accuracy of the estimate of their mean-rewards until the discarding condition is met for one of them. On the contrary, a LB pure-exploration strategy would seek to pull the arm $x \in \mathcal{X}$ whose observed reward allows to refine the estimate $\theta^*$ along the dimensions which are more suited in discriminating between the two remaining arms. Recently, the best-arm identification in linear bandits has been studied in a fixed budget setting [10], in this paper we study the sample complexity required to identify the best-linear arm with a fixed confidence.

---

[*]This work was done when the author was a visiting researcher at Microsoft Research New-England.
[†]Current affiliation: Google DeepMind.

## 2 Preliminaries

**The setting.** We consider the standard linear bandit model. Let $\mathcal{X} \subseteq \mathbb{R}^d$ be a finite set of arms, where $|\mathcal{X}| = K$ and the $\ell_2$-norm of any arm $x \in \mathcal{X}$, denoted by $||x||$, is upper-bounded by $L$. Given an unknown parameter $\theta^* \in \mathbb{R}^d$, we assume that each time an arm $x \in \mathcal{X}$ is pulled, a random reward $r(x)$ is generated according to the linear model $r(x) = x^\top \theta^* + \varepsilon$, where $\varepsilon$ is a zero-mean i.i.d. noise bounded in $[-\sigma; \sigma]$. Arms are evaluated according to their expected reward $x^\top \theta^*$ and we denote by $x^* = \arg\max_{x \in \mathcal{X}} x^\top \theta^*$ the best arm in $\mathcal{X}$. Also, we use $\Pi(\theta) = \arg\max_{x \in \mathcal{X}} x^\top \theta$ to refer to the best arm corresponding to an arbitrary parameter $\theta$. Let $\Delta(x, x') = (x - x')^\top \theta^*$ be the value *gap* between two arms, then we denote by $\Delta(x) = \Delta(x^*, x)$ the gap of $x$ w.r.t. the optimal arm and by $\Delta_{\min} = \min_{x \in \mathcal{X}} \Delta(x)$ the minimum gap, where $\Delta_{\min} > 0$. We also introduce the sets $\mathcal{Y} = \{y = x - x', \forall x, x' \in \mathcal{X}\}$ and $\mathcal{Y}^* = \{y = x^* - x, \forall x \in \mathcal{X}\}$ containing all the directions obtained as the difference of two arms (or an arm and the optimal arm) and we redefine accordingly the gap of a direction as $\Delta(y) = \Delta(x, x')$ whenever $y = x - x'$.

**The problem.** We study the best-arm identification problem. Let $\hat{x}(n)$ be the estimated best arm returned by a bandit algorithm after $n$ steps. We evaluate the *quality* of $\hat{x}(n)$ by the simple regret $R_n = (x^* - \hat{x}(n))^\top \theta^*$. While different settings can be defined (see [8] for an overview), here we focus on the $(\epsilon, \delta)$-best-arm identification problem (the so-called PAC setting), where given $\epsilon$ and $\delta \in (0, 1)$, the objective is to design an allocation strategy and a stopping criterion so that when the algorithm stops, the returned arm $\hat{x}(n)$ is such that $\mathbb{P}(R_n \geq \epsilon) \leq \delta$, while minimizing the needed number of steps. More specifically, we will focus on the case of $\epsilon = 0$ and we will provide high-probability bounds on the sample complexity $n$.

**The multi-armed bandit case.** In MAB, the complexity of best-arm identification is characterized by the gaps between arm values, following the intuition that the more similar the arms, the more pulls are needed to distinguish between them. More formally, the complexity is given by the problem-dependent quantity $H_{\text{MAB}} = \sum_{i=1}^{K} \frac{1}{\Delta_i^2}$ i.e., the inverse of the pairwise gaps between the best arm and the suboptimal arms. In the fixed budget case, $H_{\text{MAB}}$ determines the probability of returning the wrong arm [2], while in the fixed confidence case, it characterizes the sample complexity [7].

**Technical tools.** Unlike in the multi-arm bandit scenario where pulling one arm does not provide any information about other arms, in a linear model we can leverage the rewards observed over time to estimate the expected reward of all the arms in $\mathcal{X}$. Let $\mathbf{x}_n = (x_1, \ldots, x_n) \in \mathcal{X}^n$ be a sequence of arms and $(r_1, \ldots, r_n)$ the corresponding observed (random) rewards. An unbiased estimate of $\theta^*$ can be obtained by ordinary least-squares (OLS) as $\hat{\theta}_n = A_{\mathbf{x}_n}^{-1} b_{\mathbf{x}_n}$, where $A_{\mathbf{x}_n} = \sum_{t=1}^{n} x_t x_t^\top \in \mathbb{R}^{d \times d}$ and $b_{\mathbf{x}_n} = \sum_{t=1}^{n} x_t r_t \in \mathbb{R}^d$. For any fixed sequence $\mathbf{x}_n$, through Azuma's inequality, the prediction error of the OLS estimate is upper-bounded in high-probability as follows.

**Proposition 1.** *Let $c = 2\sigma\sqrt{2}$ and $c' = 6/\pi^2$. For every fixed sequence $\mathbf{x}_n$, we have[1]*

$$\mathbb{P}\left(\forall n \in \mathbb{N}, \forall x \in \mathcal{X}, |x^\top \theta^* - x^\top \hat{\theta}_n| \leq c||x||_{A_{\mathbf{x}_n}^{-1}} \sqrt{\log(c'n^2K/\delta)}\right) \geq 1 - \delta. \qquad (1)$$

While in the previous statement $\mathbf{x}_n$ is fixed, a bandit algorithm adapts the allocation in response to the rewards observed over time. In this case a different high-probability bound is needed.

**Proposition 2** (Thm. 2 in [1]). *Let $\hat{\theta}_n^\eta$ be the solution to the regularized least-squares problem with regularizer $\eta$ and let $\widetilde{A}_{\mathbf{x}}^\eta = \eta I_d + A_{\mathbf{x}}$. Then for all $x \in \mathcal{X}$ and every adaptive sequence $\mathbf{x}_n$ such that at any step $t$, $x_t$ only depends on $(x_1, r_1, \ldots, x_{t-1}, r_{t-1})$, w.p. $1 - \delta$, we have*

$$|x^\top \theta^* - x^\top \hat{\theta}_n^\eta| \leq ||x||_{(\widetilde{A}_{\mathbf{x}_n}^\eta)^{-1}} \left(\sigma\sqrt{d \log\left(\frac{1 + nL^2/\eta}{\delta}\right)} + \eta^{1/2}||\theta^*||\right). \qquad (2)$$

The crucial difference w.r.t. Eq. 1 is an additional factor $\sqrt{d}$, the price to pay for adapting $\mathbf{x}_n$ to the samples. In the sequel we will often resort to the notion of design (or "soft" allocation) $\lambda \in \mathcal{D}^k$, which prescribes the *proportions* of pulls to arm $x$ and $\mathcal{D}^k$ denotes the simplex $\mathcal{X}$. The counterpart of the design matrix $A$ for a design $\lambda$ is the matrix $\Lambda_\lambda = \sum_{x \in \mathcal{X}} \lambda(x) x x^\top$. From an allocation $\mathbf{x}_n$ we can derive the corresponding design $\lambda_{\mathbf{x}_n}$ as $\lambda_{\mathbf{x}_n}(x) = T_n(x)/n$, where $T_n(x)$ is the number of times arm $x$ is selected in $\mathbf{x}_n$, and the corresponding design matrix is $A_{\mathbf{x}_n} = n\Lambda_{\lambda_{\mathbf{x}_n}}$.

# 3 The Complexity of the Linear Best-Arm Identification Problem

As reviewed in Sect. 2, in the MAB case the complexity of the best-arm identification task is characterized by the reward gaps between the optimal and suboptimal arms. In this section, we propose an extension of the notion of complexity to the case of linear best-arm identification. In particular, we characterize the complexity by the performance of an *oracle* with access to the parameter $\theta^*$.

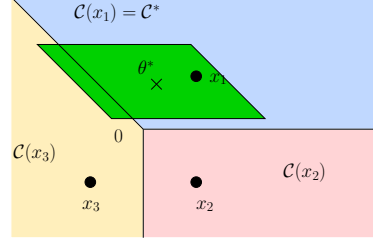

**Stopping condition.** Let $\mathcal{C}(x) = \{\theta \in \mathbb{R}^d, x \in \Pi(\theta)\}$ be the set of parameters $\theta$ which admit $x$ as an optimal arm. As illustrated in Fig. 1, $\mathcal{C}(x)$ is the cone defined by the intersection of half-spaces such that $\mathcal{C}(x) = \cap_{x' \in \mathcal{X}} \{\theta \in \mathbb{R}^d, (x - x')^\top \theta \geq 0\}$ and all the cones together form a partition of the Euclidean space $\mathbb{R}^d$. We assume that the oracle knows the cone $\mathcal{C}(x^*)$ containing all the parameters for which $x^*$ is optimal. Furthermore, we assume

Figure 1: The cones corresponding to three arms (dots) in $\mathbb{R}^2$. Since $\theta^* \in \mathcal{C}(x_1)$, then $x^* = x_1$. The confidence set $\mathcal{S}^*(\mathbf{x}_n)$ (in green) is aligned with directions $x_1 - x_2$ and $x_1 - x_3$. Given the uncertainty in $\mathcal{S}^*(\mathbf{x}_n)$, both $x_1$ and $x_3$ may be optimal.

that for any allocation $\mathbf{x}_n$, it is possible to construct a confidence set $\mathcal{S}^*(\mathbf{x}_n) \subseteq \mathbb{R}^d$ such that $\theta^* \in \mathcal{S}^*(\mathbf{x}_n)$ and the (random) OLS estimate $\hat{\theta}_n$ belongs to $\mathcal{S}^*(\mathbf{x}_n)$ with high probability, i.e., $\mathbb{P}(\hat{\theta}_n \in \mathcal{S}^*(\mathbf{x}_n)) \geq 1 - \delta$. As a result, the oracle stopping criterion simply checks whether the confidence set $\mathcal{S}^*(\mathbf{x}_n)$ is contained in $\mathcal{C}(x^*)$ or not. In fact, whenever for an allocation $\mathbf{x}_n$ the set $\mathcal{S}^*(\mathbf{x}_n)$ overlaps the cones of different arms $x \in \mathcal{X}$, there is ambiguity in the identity of the arm $\Pi(\hat{\theta}_n)$. On the other hand when all possible values of $\hat{\theta}_n$ are included with high probability in the "right" cone $C(x^*)$, then the optimal arm is returned.

**Lemma 1.** *Let $\mathbf{x}_n$ be an allocation such that $\mathcal{S}^*(\mathbf{x}_n) \subseteq \mathcal{C}(x^*)$. Then $\mathbb{P}(\Pi(\hat{\theta}_n) \neq x^*) \leq \delta$.*

**Arm selection strategy.** From the previous lemma[2] it follows that the objective of an arm selection strategy is to define an allocation $\mathbf{x}_n$ which leads to $\mathcal{S}^*(\mathbf{x}_n) \subseteq \mathcal{C}(x^*)$ as quickly as possible.[3] Since this condition only depends on deterministic objects ($\mathcal{S}^*(\mathbf{x}_n)$ and $\mathcal{C}(x^*)$), it can be computed independently from the actual reward realizations. From a geometrical point of view, this corresponds to choosing arms so that the confidence set $\mathcal{S}^*(\mathbf{x}_n)$ shrinks into the optimal cone $\mathcal{C}(x^*)$ within the smallest number of pulls. To characterize this strategy we need to make explicit the form of $\mathcal{S}^*(\mathbf{x}_n)$. Intuitively speaking, the more $\mathcal{S}^*(\mathbf{x}_n)$ is "aligned" with the boundaries of the cone, the easier it is to shrink it into the cone. More formally, the condition $\mathcal{S}^*(\mathbf{x}_n) \subseteq \mathcal{C}(x^*)$ is equivalent to

$$\forall x \in \mathcal{X}, \forall \theta \in \mathcal{S}^*(\mathbf{x}_n), (x^* - x)^\top \theta \geq 0 \iff \forall y \in \mathcal{Y}^*, \forall \theta \in \mathcal{S}^*(\mathbf{x}_n), y^\top(\theta^* - \theta) \leq \Delta(y).$$

Then we can simply use Prop. 1 to directly control the term $y^\top(\theta^* - \theta)$ and define

$$\mathcal{S}^*(\mathbf{x}_n) = \left\{ \theta \in \mathbb{R}^d, \forall y \in \mathcal{Y}^*, y^\top(\theta^* - \theta) \leq c\|y\|_{A_{\mathbf{x}_n}^{-1}} \sqrt{\log_n(K^2/\delta)} \right\}. \tag{3}$$

Thus the stopping condition $\mathcal{S}^*(\mathbf{x}_n) \subseteq \mathcal{C}(x^*)$ is equivalent to the condition that, for any $y \in \mathcal{Y}^*$,

$$c\|y\|_{A_{\mathbf{x}_n}^{-1}} \sqrt{\log_n(K^2/\delta)} \leq \Delta(y). \tag{4}$$

From this condition, the oracle allocation strategy simply follows as

$$\mathbf{x}_n^* = \arg\min_{\mathbf{x}_n} \max_{y \in \mathcal{Y}^*} \frac{c\|y\|_{A_{\mathbf{x}_n}^{-1}} \sqrt{\log_n(K^2/\delta)}}{\Delta(y)} = \arg\min_{\mathbf{x}_n} \max_{y \in \mathcal{Y}^*} \frac{\|y\|_{A_{\mathbf{x}_n}^{-1}}}{\Delta(y)}. \tag{5}$$

Notice that this strategy does not return an uniformly accurate estimate of $\theta^*$ but it rather pulls arms that allow to reduce the uncertainty of the estimation of $\theta^*$ over the directions of interest (i.e., $\mathcal{Y}^*$) below their corresponding gaps. This implies that the objective of Eq. 5 is to exploit the global linear assumption by pulling any arm in $\mathcal{X}$ that could give information about $\theta^*$ over the directions in $\mathcal{Y}^*$, so that directions with small gaps are better estimated than those with bigger gaps.

**Sample complexity.** We are now ready to define the sample complexity of the oracle, which corresponds to the minimum number of steps needed by the allocation in Eq. 5 to achieve the stopping condition in Eq. 4. From a technical point of view, it is more convenient to express the complexity of the problem in terms of the optimal design (soft allocation) instead of the discrete allocation $\mathbf{x}_n$. Let $\rho^*(\lambda) = \max_{y \in \mathcal{Y}^*} ||y||^2_{\Lambda_\lambda^{-1}} / \Delta^2(y)$ be the square of the objective function in Eq. 5 for any design $\lambda \in \mathcal{D}^k$. We define the complexity of a linear best-arm identification problem as the performance achieved by the optimal design $\lambda^* = \arg\min_\lambda \rho^*(\lambda)$, i.e.

$$H_{\text{LB}} = \min_{\lambda \in \mathcal{D}^k} \max_{y \in \mathcal{Y}^*} \frac{||y||^2_{\Lambda_\lambda^{-1}}}{\Delta^2(y)} = \rho^*(\lambda^*). \tag{6}$$

This definition of complexity is less explicit than in the case of $H_{\text{MAB}}$ but it contains similar elements, notably the inverse of the gaps squared. Nonetheless, instead of summing the inverses over all the arms, $H_{\text{LB}}$ implicitly takes into consideration the correlation between the arms in the term $||y||^2_{\Lambda_\lambda^{-1}}$, which represents the uncertainty in the estimation of the gap between $x^*$ and $x$ (when $y = x^* - x$). As a result, from Eq. 4 the sample complexity becomes

$$N^* = c^2 H_{\text{LB}} \log_n(K^2/\delta), \tag{7}$$

where we use the fact that, if implemented over $n$ steps, $\lambda^*$ induces a design matrix $A_{\lambda^*} = n\Lambda_{\lambda^*}$ and $\max_y ||y||^2_{A_{\lambda^*}^{-1}} / \Delta^2(y) = \rho^*(\lambda^*)/n$. Finally, we bound the range of the complexity.

**Lemma 2.** *Given an arm set $\mathcal{X} \subseteq \mathbb{R}^d$ and a parameter $\theta^*$, the complexity $H_{LB}$ (Eq. 6) is such that*

$$\max_{y \in \mathcal{Y}^*} ||y||^2/(L\Delta^2_{\min}) \leq H_{LB} \leq 4d/\Delta^2_{\min}. \tag{8}$$

*Furthermore, if $\mathcal{X}$ is the canonical basis, the problem reduces to a MAB and $H_{MAB} \leq H_{LB} \leq 2H_{MAB}$.*

The previous bounds show that $\Delta_{\min}$ plays a significant role in defining the complexity of the problem, while the specific shape of $\mathcal{X}$ impacts the numerator in different ways. In the worst case the full dimensionality $d$ appears (upper-bound), and more arm-set specific quantities, such as the norm of the arms $L$ and of the directions $\mathcal{Y}^*$, appear in the lower-bound.

## 4 Static Allocation Strategies

The oracle stopping condition (Eq. 4) and allocation strategy (Eq. 5) cannot be implemented in practice since $\theta^*$, the gaps $\Delta(y)$, and the directions $\mathcal{Y}^*$ are unknown. In this section we investigate how to define algorithms that only rely on the information available from $\mathcal{X}$ and the samples collected over time. We introduce an empirical stopping criterion and two static allocations.

**Empirical stopping criterion.** The stopping condition $\mathcal{S}^*(\mathbf{x}_n) \subseteq \mathcal{C}(x^*)$ cannot be tested since $\mathcal{S}^*(\mathbf{x}_n)$ is centered in the unknown parameter $\theta^*$ and $\mathcal{C}(x^*)$ depends on the unknown optimal arm $x^*$. Nonetheless, we notice that given $\mathcal{X}$, for each

> **Input:** decision space $\mathcal{X} \in \mathbb{R}^d$, confidence $\delta > 0$
> Set: $t = 0$; $Y = \{y = (x - x'); x \neq x' \in \mathcal{X}\}$;
> **while** Eq. 11 is not true **do**
>   **if** $G$-allocation **then**
>     $x_t = \arg\min_{x \in X} \max_{x' \in X} x'^\top (A + xx^\top)^{-1} x'$
>   **else if** $\mathcal{X}\mathcal{Y}$-allocation **then**
>     $x_t = \arg\min_{x \in X} \max_{y \in Y} y^\top (A + xx^\top)^{-1} y$
>   **end if**
>   Update $\hat{\theta}_t = A_t^{-1} b_t, t = t + 1$
> **end while**
> Return arm $\Pi(\hat{\theta}_t)$

Figure 2: Static allocation algorithms

$x \in \mathcal{X}$ the cones $\mathcal{C}(x)$ can be constructed beforehand. Let $\widehat{\mathcal{S}}(\mathbf{x}_n)$ be a high-probability confidence set such that for any $\mathbf{x}_n$, $\hat{\theta}_n \in \widehat{\mathcal{S}}(\mathbf{x}_n)$ and $\mathbb{P}(\theta^* \in \widehat{\mathcal{S}}(\mathbf{x}_n)) \geq 1 - \delta$. Unlike $\mathcal{S}^*$, $\widehat{\mathcal{S}}$ can be directly computed from samples and we can stop whenever there exists an $x$ such that $\widehat{\mathcal{S}}(\mathbf{x}_n) \subseteq \mathcal{C}(x)$.

**Lemma 3.** *Let $\mathbf{x}_n = (x_1, \ldots, x_n)$ be an arbitrary allocation sequence. If after $n$ steps there exists an arm $x \in \mathcal{X}$ such that $\widehat{\mathcal{S}}(\mathbf{x}_n) \subseteq \mathcal{C}(x)$ then $\mathbb{P}(\Pi(\hat{\theta}_n) \neq x^*) \leq \delta$.*

**Arm selection strategy.** Similarly to the oracle algorithm, we should design an allocation strategy that guarantees that the (random) confidence set $\widehat{\mathcal{S}}(\mathbf{x}_n)$ shrinks in one of the cones $\mathcal{C}(x)$ within the fewest number of steps. Let $\widehat{\Delta}_n(x, x') = (x - x')^\top \hat{\theta}_n$ be the empirical gap between arms $x, x'$. Then the stopping condition $\widehat{\mathcal{S}}(\mathbf{x}_n) \subseteq \mathcal{C}(x)$ can be written as

$$\exists x \in \mathcal{X}, \forall x' \in \mathcal{X}, \forall \theta \in \widehat{\mathcal{S}}(\mathbf{x}_n), (x - x')^\top \theta \geq 0$$
$$\Leftrightarrow \quad \exists x \in \mathcal{X}, \forall x' \in \mathcal{X}, \forall \theta \in \widehat{\mathcal{S}}(\mathbf{x}_n), (x - x')^\top (\hat{\theta}_n - \theta) \leq \widehat{\Delta}_n(x, x'). \tag{9}$$

This suggests that the empirical confidence set can be defined as

$$\widehat{\mathcal{S}}(\mathbf{x}_n) = \left\{ \theta \in \mathbb{R}^d, \forall y \in \mathcal{Y}, y^\top(\hat{\theta}_n - \theta) \le c\|y\|_{A_{\mathbf{x}_n}^{-1}} \sqrt{\log_n(K^2/\delta)} \right\}. \tag{10}$$

Unlike $\mathcal{S}^*(\mathbf{x}_n)$, $\widehat{\mathcal{S}}(\mathbf{x}_n)$ is centered in $\hat{\theta}_n$ and it considers all directions $y \in \mathcal{Y}$. As a result, the stopping condition in Eq. 9 could be reformulated as

$$\exists x \in \mathcal{X}, \forall x' \in \mathcal{X}, c\|x - x'\|_{A_{\mathbf{x}_n}^{-1}} \sqrt{\log_n(K^2/\delta)} \le \widehat{\Delta}_n(x, x'). \tag{11}$$

Although similar to Eq. 4, unfortunately this condition cannot be directly used to derive an allocation strategy. In fact, it is considerably more difficult to define a suitable allocation strategy to fit a random confidence set $\widehat{\mathcal{S}}$ into a cone $\mathcal{C}(x)$ for an $x$ which is not known in advance. In the following we propose two allocations that try to achieve the condition in Eq. 11 as fast as possible by implementing a static arm selection strategy, while we present a more sophisticated adaptive strategy in Sect. 5. The general structure of the static allocations in summarized in Fig. 2.

**G-Allocation Strategy.** The definition of the $G$-allocation strategy directly follows from the observation that for any pair $(x, x') \in \mathcal{X}^2$ we have that $\|x - x'\|_{A_{\mathbf{x}_n}^{-1}} \le 2 \max_{x'' \in \mathcal{X}} \|x''\|_{A_{\mathbf{x}_n}^{-1}}$. This suggests that an allocation minimizing $\max_{x \in \mathcal{X}} \|x\|_{A_{\mathbf{x}_n}^{-1}}$ reduces an upper bound on the quantity tested in the stopping condition in Eq. 11. Thus, for any fixed $n$, we define the $G$-allocation as

$$\mathbf{x}_n^G = \arg\min_{\mathbf{x}_n} \max_{x \in \mathcal{X}} \|x\|_{A_{\mathbf{x}_n}^{-1}}. \tag{12}$$

We notice that this formulation coincides with the standard $G$-optimal design (hence the name of the allocation) defined in experimental design theory [15, Sect. 9.2] to minimize the maximal mean-squared prediction error in linear regression. The $G$-allocation can be interpreted as the design that allows to estimate $\theta^*$ *uniformly well* over all the arms in $\mathcal{X}$. Notice that the $G$-allocation in Eq. 12 is well defined only for a fixed number of steps $n$ and it cannot be directly implemented in our case, since $n$ is unknown in advance. Therefore we have to resort to a more "incremental" implementation. In the experimental design literature a wide number of approximate solutions have been proposed to solve the $NP$-hard discrete optimization problem in Eq. 12 (see [4, 17] for some recent results and [18] for a more thorough discussion). For any approximate $G$-allocation strategy with performance no worse than a factor $(1 + \beta)$ of the optimal strategy $\mathbf{x}_n^G$, the sample complexity $N^G$ is bounded as follows.

**Theorem 1.** *If the $G$-allocation strategy is implemented with a $\beta$-approximate method and the stopping condition in Eq. 11 is used, then*

$$\mathbb{P}\left[N^G \le \frac{16c^2 d(1 + \beta)\log_n(K^2/\delta)}{\Delta_{\min}^2} \wedge \Pi(\hat{\theta}_{N^G}) = x^*\right] \ge 1 - \delta. \tag{13}$$

Notice that this result matches (up to constants) the worst-case value of $N^*$ given the upper bound on $H_{\text{LB}}$. This means that, although completely static, the $G$-allocation is already worst-case optimal.

**$\mathcal{X}\mathcal{Y}$-Allocation Strategy.** Despite being worst-case optimal, $G$-allocation is minimizing a rather loose upper bound on the quantity used to test the stopping criterion. Thus, we define an alternative static allocation that targets the stopping condition in Eq. 11 more directly by reducing its left-hand-side for any possible direction in $\mathcal{Y}$. For any fixed $n$, we define the $\mathcal{X}\mathcal{Y}$-allocation as

$$\mathbf{x}_n^{\mathcal{X}\mathcal{Y}} = \arg\min_{\mathbf{x}_n} \max_{y \in \mathcal{Y}} \|y\|_{A_{\mathbf{x}_n}^{-1}}. \tag{14}$$

$\mathcal{X}\mathcal{Y}$-allocation is based on the observation that the stopping condition in Eq. 11 requires only the empirical gaps $\widehat{\Delta}(x, x')$ to be well estimated, hence arms are pulled with the objective of increasing the accuracy of directions in $\mathcal{Y}$ instead of arms $\mathcal{X}$. This problem can be seen as a transductive variant of the $G$-optimal design [19], where the target vectors $\mathcal{Y}$ are different from the vectors $\mathcal{X}$ used in the design. The sample complexity of the $\mathcal{X}\mathcal{Y}$-allocation is as follows.

**Theorem 2.** *If the $\mathcal{X}\mathcal{Y}$-allocation strategy is implemented with a $\beta$-approximate method and the stopping condition in Eq. 11 is used, then*

$$\mathbb{P}\left[N^{\mathcal{X}\mathcal{Y}} \le \frac{32c^2 d(1 + \beta)\log_n(K^2/\delta)}{\Delta_{\min}^2} \wedge \Pi(\hat{\theta}_{N^{\mathcal{X}\mathcal{Y}}}) = x^*\right] \ge 1 - \delta. \tag{15}$$

Although the previous bound suggests that $\mathcal{X}\mathcal{Y}$ achieves a performance comparable to the $G$-allocation, in fact $\mathcal{X}\mathcal{Y}$ may be arbitrarily better than $G$-allocation (for an example, see [18]).

# 5 $\mathcal{XY}$-Adaptive Allocation Strategy

**Fully adaptive allocation strategies.** Although both $G$- and $\mathcal{XY}$-allocation are sound since they minimize upper-bounds on the quantities used by the stopping condition (Eq. 11), they may be very sub-optimal w.r.t. the ideal performance of the oracle introduced in Sec. 3. Typically, an improvement can be obtained by moving to strategies adapting on the rewards observed over time. Nonetheless, as reported in Prop. 2, whenever $\mathbf{x}_n$ is not a fixed sequence, the bound in Eq. 2 should be used. As a result, a factor $\sqrt{d}$ would appear in the definition of the confidence sets and in the stopping condition. This directly implies that the sample complexity of a fully adaptive strategy would scale linearly with the dimensionality $d$ of the problem, thus removing any advantage w.r.t. static allocations. In fact, the sample complexity of $G$- and $\mathcal{XY}$-allocation already scales linearly with $d$ and from Lem. 2 we cannot expect to improve the dependency on $\Delta_{\min}$. Thus, on the one hand, we need to use the tighter bounds in Eq. 1 and, on the other hand, we require to be adaptive w.r.t. samples. In the sequel we propose a phased algorithm which successfully meets both requirements using a static allocation within each phase but choosing the type of allocation depending on the samples observed in previous phases.

---

**Input:** decision space $\mathcal{X} \in \mathbb{R}^d$; parameter $\alpha$; confidence $\delta$
Set $j = 1$; $\widehat{\mathcal{X}}_j = \mathcal{X}$; $\widehat{\mathcal{Y}}_1 = \mathcal{Y}$; $\rho_0 = 1$; $n_0 = d(d+1) + 1$
**while** $|\widehat{\mathcal{X}}_j| > 1$ **do**
$\quad \rho^j = \rho^{j-1}$
$\quad t = 1; A_0 = I$
$\quad$ **while** $\rho^j / t \geq \alpha \rho^{j-1}(\mathbf{x}_{n_{j-1}}^{j-1}) / n_{j-1}$ **do**
$\quad\quad$ Select arm $x_t = \arg\min_{x \in X} \max_{y \in Y} y^\top (A + xx^\top)^{-1} y$
$\quad\quad$ Update $A_t = A_{t-1} + x_t x_t^\top, t = t + 1$
$\quad\quad \rho^j = \max_{y \in \widehat{\mathcal{Y}}_j} y^\top A_t^{-1} y$
$\quad$ **end while**
$\quad$ Compute $b = \sum_{s=1}^t x_s r_s$; $\hat{\theta}_j = A_t^{-1} b$
$\quad \widehat{\mathcal{X}}_{j+1} = \mathcal{X}$
$\quad$ **for** $x \in \mathcal{X}$ **do**
$\quad\quad$ **if** $\exists x' : ||x - x'||_{A_t^{-1}} \sqrt{\log_n(K^2/\delta)} \leq \widehat{\Delta}_j(x', x)$ **then**
$\quad\quad\quad \widehat{\mathcal{X}}_{j+1} = \widehat{\mathcal{X}}_{j+1} - \{x\}$
$\quad\quad$ **end if**
$\quad$ **end for**
$\quad \widehat{\mathcal{Y}}_{j+1} = \{y = (x - x'); x, x' \in \widehat{\mathcal{X}}_{j+1}\}$
**end while**
Return $\Pi(\hat{\theta}_j)$

---

Figure 3: $\mathcal{XY}$-Adaptive allocation algorithm

**Algorithm.** The ideal case would be to define an empirical version of the oracle allocation in Eq. 5 so as to adjust the accuracy of the prediction only on the directions of interest $\mathcal{Y}^*$ and according to their gaps $\Delta(y)$. As discussed in Sect. 4 this cannot be obtained by a direct adaptation of Eq. 11. In the following, we describe a safe alternative to adjust the allocation strategy to the gaps.

**Lemma 4.** *Let $\mathbf{x}_n$ be a fixed allocation sequence and $\hat{\theta}_n$ its corresponding estimate for $\theta^*$. If an arm $x \in \mathcal{X}$ is such that*

$$\exists x' \in \mathcal{X} \text{ s.t. } c||x' - x||_{A_{\mathbf{x}_n}^{-1}} \sqrt{\log_n(K^2/\delta)} < \widehat{\Delta}_n(x', x), \tag{16}$$

*then arm $x$ is sub-optimal. Moreover, if Eq. 16 is true, we say that $x'$ dominates $x$.*

Lem. 4 allows to easily construct the set of potentially optimal arms, denoted $\widehat{\mathcal{X}}(\mathbf{x}_n)$, by removing from $\mathcal{X}$ all the dominated arms. As a result, we can replace the stopping condition in Eq. 11, by just testing whether the number of non-dominated arms $|\widehat{\mathcal{X}}(\mathbf{x}_n)|$ is equal to 1, which corresponds to the case where the confidence set is fully contained into a single cone. Using $\widehat{\mathcal{X}}(\mathbf{x}_n)$, we construct $\widehat{\mathcal{Y}}(\mathbf{x}_n) = \{y = x - x'; x, x' \in \widehat{\mathcal{X}}(\mathbf{x}_n)\}$, the set of directions along which the estimation of $\theta^*$ needs to be improved to further shrink $\widehat{\mathcal{S}}(\mathbf{x}_n)$ into a single cone and trigger the stopping condition. Note that if $\mathbf{x}_n$ was an adaptive strategy, then we could not use Lem. 4 to discard arms but we should rely on the bound in Prop. 2. To avoid this problem, an effective solution is to run the algorithm through phases. Let $j \in \mathbb{N}$ be the index of a phase and $n_j$ its corresponding length. We denote by $\widehat{\mathcal{X}}_j$ the set of non-dominated arms constructed on the basis of the samples collected in the phase $j - 1$. This set is used to identify the directions $\widehat{\mathcal{Y}}_j$ and to define a *static* allocation which focuses on reducing the uncertainty of $\theta^*$ along the directions in $\widehat{\mathcal{Y}}_j$. Formally, in phase $j$ we implement the allocation

$$\mathbf{x}_{n_j}^j = \arg\min_{\mathbf{x}_{n_j}} \max_{y \in \widehat{\mathcal{Y}}_j} ||y||_{A_{\mathbf{x}_{n_j}}^{-1}}, \tag{17}$$

which coincides with a $\mathcal{XY}$-allocation (see Eq. 14) but restricted on $\widehat{\mathcal{Y}}_j$. Notice that $\mathbf{x}_{n_j}^j$ may still use any arm in $\mathcal{X}$ which could be useful in reducing the confidence set along any of the directions in

$\widehat{\mathcal{Y}}_j$. Once phase $j$ is over, the OLS estimate $\hat{\theta}^j$ is computed using the rewards observed within phase $j$ and then is used to test the stopping condition in Eq. 11. Whenever the stopping condition does not hold, a new set $\widehat{\mathcal{X}}_{j+1}$ is constructed using the discarding condition in Lem. 4 and a new phase is started. Notice that through this process, at each phase $j$, the allocation $\mathbf{x}_{n_j}^j$ is static conditioned on the previous allocations and the use of the bound from Prop. 1 is still correct.

A crucial aspect of this algorithm is the length of the phases $n_j$. On the one hand, short phases allow a high rate of adaptivity, since $\widehat{\mathcal{X}}_j$ is recomputed very often. On the other hand, if a phase is too short, it is very unlikely that the estimate $\hat{\theta}^j$ may be accurate enough to actually discard any arm. An effective way to define the length of a phase in a deterministic way is to relate it to the actual uncertainty of the allocation in estimating the value of all the active directions in $\widehat{\mathcal{Y}}_j$. In phase $j$, let $\rho^j(\lambda) = \max_{y \in \widehat{\mathcal{Y}}_j} ||y||^2_{\Lambda_\lambda^{-1}}$, then given a parameter $\alpha \in (0, 1)$, we define

$$n_j = \min \left\{ n \in \mathbb{N} : \rho^j(\lambda_{\mathbf{x}_n^j})/n \le \alpha \rho^{j-1}(\lambda^{j-1})/n_{j-1} \right\}, \tag{18}$$

where $\mathbf{x}_n^j$ is the allocation defined in Eq. 17 and $\lambda^{j-1}$ is the design corresponding to $\mathbf{x}_{n_{j-1}}^{j-1}$, the allocation performed at phase $j - 1$. In words, $n_j$ is the minimum number of steps needed by the $\mathcal{X}\mathcal{Y}$-adaptive allocation to achieve an uncertainty over all the directions of interest which is a fraction $\alpha$ of the performance obtained in the previous iteration. Notice that given $\widehat{\mathcal{Y}}_j$ and $\rho^{j-1}$ this quantity can be computed before the actual beginning of phase $j$. The resulting algorithm using the $\mathcal{X}\mathcal{Y}$-Adaptive allocation strategy is summarized in Fig. 3.

**Sample complexity.** Although the $\mathcal{X}\mathcal{Y}$-Adaptive allocation strategy is designed to approach the oracle sample complexity $N^*$, in early phases it basically implements a $\mathcal{X}\mathcal{Y}$-allocation and no significant improvement can be expected until some directions are discarded from $\widehat{\mathcal{Y}}$. At that point, $\mathcal{X}\mathcal{Y}$-adaptive starts focusing on directions which only contain near-optimal arms and it starts approaching the behavior of the oracle. As a result, in studying the sample complexity of $\mathcal{X}\mathcal{Y}$-Adaptive we have to take into consideration the unavoidable price of discarding "suboptimal" directions. This cost is directly related to the geometry of the arm space that influences the number of samples needed before arms can be discarded from $\mathcal{X}$. To take into account this problem-dependent quantity, we introduce a slightly relaxed definition of complexity. More precisely, we define the number of steps needed to discard all the directions which do not contain $x^*$, i.e. $\mathcal{Y} - \mathcal{Y}^*$. From a geometrical point of view, this corresponds to the case when for any pair of suboptimal arms $(x, x')$, the confidence set $\mathcal{S}^*(\mathbf{x}_n)$ does not intersect the hyperplane separating the cones $\mathcal{C}(x)$ and $\mathcal{C}(x')$. Fig. 1 offers a simple illustration for such a situation: $\mathcal{S}^*$ no longer intercepts the border line between $\mathcal{C}(x_2)$ and $\mathcal{C}(x_3)$, which implies that direction $x_2 - x_3$ can be discarded. More formally, the hyperplane containing parameters $\theta$ for which $x$ and $x'$ are equivalent is simply $\mathcal{C}(x) \cap \mathcal{C}(x')$ and the quantity

$$M^* = \min\{n \in \mathbb{N}, \forall x \ne x^*, \forall x' \ne x^*, \mathcal{S}^*(\mathbf{x}_n^{\mathcal{X}\mathcal{Y}}) \cap (\mathcal{C}(x) \cap \mathcal{C}(x')) = \emptyset\} \tag{19}$$

corresponds to the minimum number of steps needed by the static $\mathcal{X}\mathcal{Y}$-allocation strategy to discard all the *suboptimal* directions. This term together with the oracle complexity $N^*$ characterizes the sample complexity of the phases of the $\mathcal{X}\mathcal{Y}$-adaptive allocation. In fact, the length of the phases is such that either they correspond to the complexity of the oracle or they can never last more than the steps needed to discard all the sub-optimal directions. As a result, the overall sample complexity of the $\mathcal{X}\mathcal{Y}$-adaptive algorithm is bounded as in the following theorem.

**Theorem 3.** *If the $\mathcal{X}\mathcal{Y}$-Adaptive allocation strategy is implemented with a $\beta$-approximate method and the stopping condition in Eq. 11 is used, then*

$$\mathbb{P}\left[ N \le \frac{(1+\beta)\max\{M^*, \frac{16}{\alpha}N^*\}}{\log(1/\alpha)} \log\left(\frac{c\sqrt{\log_n(K^2/\delta)}}{\Delta_{\min}}\right) \wedge \Pi(\hat{\theta}_N) = x^* \right] \ge 1 - \delta. \tag{20}$$

We first remark that, unlike $G$ and $\mathcal{X}\mathcal{Y}$, the sample complexity of $\mathcal{X}\mathcal{Y}$-Adaptive does not have any direct dependency on $d$ and $\Delta_{\min}$ (except in the logarithmic term) but it rather scales with the oracle complexity $N^*$ and the cost of discarding suboptimal directions $M^*$. Although this additional cost is probably unavoidable, one may have expected that $\mathcal{X}\mathcal{Y}$-Adaptive may need to discard all the suboptimal directions before performing as well as the oracle, thus having a sample complexity of $O(M^* + N^*)$. Instead, we notice that $N$ scales with the *maximum* of $M^*$ and $N^*$, thus implying that $\mathcal{X}\mathcal{Y}$-Adaptive may actually catch up with the performance of the oracle (with only a multiplicative factor of $16/\alpha$) whenever discarding suboptimal directions is less expensive than actually identifying the best arm.

## 6 Numerical Simulations

We illustrate the performance of $\mathcal{XY}$-Adaptive and compare it to the $\mathcal{XY}$-Oracle strategy (Eq. 5), the static allocations $\mathcal{XY}$ and $G$, as well as with the fully-adaptive version of $\mathcal{XY}$ where $\widehat{\mathcal{X}}$ is updated at each round and the bound from Prop.2 is used. For a fixed confidence $\delta = 0.05$, we compare the sampling budget needed to identify the best arm with probability at least $1 - \delta$. We consider a set of arms $\mathcal{X} \in \mathbb{R}^d$, with $|\mathcal{X}| = d + 1$ including the canonical basis $(e_1, \ldots, e_d)$ and an additional arm $x_{d+1} = [\cos(\omega) \;\; \sin(\omega) \;\; 0 \;\; \ldots \;\; 0]^\top$. We choose $\theta^* = [2 \;\; 0 \;\; 0 \;\; \ldots \;\; 0]^\top$, and fix $\omega = 0.01$, so that $\Delta_{\min} = (x_1 - x_{d+1})^\top \theta^*$ is much smaller than the other gaps. In this setting, an efficient sampling strategy should focus on reducing the uncertainty in the direction $\tilde{y} = (x_1 - x_{d+1})$ by pulling the arm $x_2 = e_2$ which is almost aligned with $\tilde{y}$. In fact, from the rewards obtained from $x_2$ it is easier to decrease the uncertainty about the second component of $\theta^*$, that is precisely the dimension which allows to discriminate between $x_1$ and $x_{d+1}$. Also, we fix $\alpha = 1/10$, and the noise $\varepsilon \sim \mathcal{N}(0, 1)$. Each phase begins with an initialization matrix $A_0$, obtained by pulling once each canonical arm. In Fig. 4 we report the sampling budget of the algorithms, averaged over 100 runs, for $d = 2 \ldots 10$.

**The results.** The numerical results show that $\mathcal{XY}$-Adaptive is effective in allocating the samples to shrink the uncertainty in the direction $\tilde{y}$. Indeed, $\mathcal{XY}$-adaptive identifies the most important direction after few phases and is able to perform an allocation which mimics that of the oracle. On the contrary, $\mathcal{XY}$ and $G$ do not adjust to the empirical gaps and consider all directions as equally important. This behavior forces $\mathcal{XY}$ and $G$ to allocate samples until the uncertainty is smaller than $\Delta_{\min}$ in all directions. Even though the Fully-adaptive algorithm also identifies the most informative direction rapidly, the $\sqrt{d}$ term in the bound delays the discarding of the arms and prevents the algorithm from gaining any advantage compared to $\mathcal{XY}$ and $G$. As shown in Fig. 4,

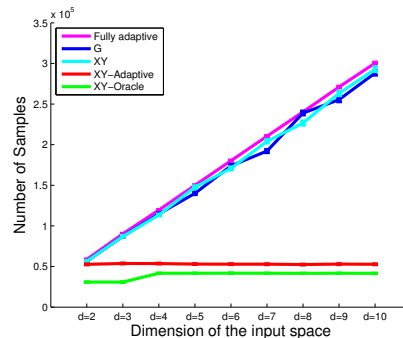

Figure 4: The sampling budget needed to identify the best arm, when the dimension grows from $\mathbb{R}^2$ to $\mathbb{R}^{10}$.

the difference between the budget of $\mathcal{XY}$-Adaptive and the static strategies increases with the number of dimensions. In fact, while additional dimensions have little to no impact on $\mathcal{XY}$-Oracle and $\mathcal{XY}$-Adaptive (the only important direction remains $\tilde{y}$ independently from the number of unknown features of $\theta^*$), for the static allocations more dimensions imply more directions to be considered and more features of $\theta^*$ to be estimated uniformly well until the uncertainty falls below $\Delta_{\min}$.

## 7 Conclusions

In this paper we studied the problem of best-arm identification with a fixed confidence, in the linear bandit setting. First we offered a preliminary characterization of the problem-dependent complexity of the best arm identification task and shown its connection with the complexity in the MAB setting. Then, we designed and analyzed efficient sampling strategies for this problem. The $G$-allocation strategy allowed us to point out a close connection with optimal experimental design techniques, and in particular to the G-optimality criterion. Through the second proposed strategy, $\mathcal{XY}$-allocation, we introduced a novel optimal design problem where the testing arms do not coincide with the arms chosen in the design. Lastly, we pointed out the limits that a fully-adaptive allocation strategy might have in the linear bandit setting and proposed a phased-algorithm, $\mathcal{XY}$-Adaptive, that learns from previous observations, without suffering from the dimensionality of the problem. Since this is one of the first works that analyze pure-exploration problems in the linear-bandit setting, it opens the way for an important number of similar problems already studied in the MAB setting. For instance, we can investigate strategies to identify the best-linear arm when having a limited budget or study the best-arm identification when the set of arms is very large (or infinite). Some interesting extensions also emerge from the optimal experimental design literature, such as the study of sampling strategies for meeting the G-optimality criterion when the noise is heterosckedastic, or the design of efficient strategies for satisfying other related optimality criteria, such as V-optimality.

**Acknowledgments** This work was supported by the French Ministry of Higher Education and Research, Nord-Pas de Calais Regional Council and FEDER through the "Contrat de Projets Etat Region 2007–2013", and European Community's Seventh Framework Programme under grant agreement no 270327 (project CompLACS).

## Footnotes

[1] Whenever Prop.1 is used for all directions $y \in \mathcal{Y}$, then the logarithmic term becomes $\log(c'n^2K^2/\delta)$ because of an additional union bound. For the sake of simplicity, in the sequel we always use $\log_n(K^2/\delta)$.

[2]For all the proofs in this paper, we refer the reader to the long version of the paper [18].

[3]Notice that by definition of the confidence set and since $\theta_n \to \theta^*$ as $n \to \infty$, any strategy repeatedly pulling all the arms would eventually meet the stopping condition.

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
