[Supplementary Material]

## A Comparison between G-allocation and $\mathcal{XY}$-allocation

We define two examples illustrating the difference between the $G$ and the $\mathcal{XY}$ allocation strategies. Let us consider a problem with $\mathcal{X} \subset \mathbb{R}^2$ and arms $x_1 = [1 \ \epsilon/2]^\top$ and $x_2 = [1 \ -\epsilon/2]^\top$, where $\epsilon \in (0, 1)$. In this case, both static allocations pull the two arms the same number of times, thus inducing an optimal design $\lambda(x_1) = \lambda(x_2) = 1/2$. We want to study the (asymptotic) performance of the allocation according to the different definition of error $\max_{x \in \mathcal{X}} x^\top \Lambda_\lambda^{-1} x$ and $\max_{y \in \mathcal{Y}} y^\top \Lambda_\lambda^{-1} y$ used by $G$ and $\mathcal{XY}$-allocation respectively. We first notice that

$$\Lambda_\lambda = \frac{1}{2} \begin{bmatrix} 1 & \epsilon/2 \\ \epsilon/2 & \epsilon^2/4 \end{bmatrix} + \frac{1}{2} \begin{bmatrix} 1 & -\epsilon/2 \\ -\epsilon/2 & \epsilon^2/4 \end{bmatrix} = \begin{bmatrix} 1 & 0 \\ 0 & \epsilon^2/4 \end{bmatrix}.$$

As a result, for both $x_1$ and $x_2$ we have

$$[1 \ \epsilon/2]\Lambda_\lambda^{-1} \begin{bmatrix} 1 \\ \epsilon/2 \end{bmatrix} = [1 \ \epsilon/2] \begin{bmatrix} 1 & 0 \\ 0 & 4/\epsilon^2 \end{bmatrix} \begin{bmatrix} 1 \\ \epsilon/2 \end{bmatrix} = 2.$$

On the other hand, if we consider the direction $y = x_1 - x_2 = [0 \ \epsilon]$, we have

$$[0 \ \epsilon]\Lambda_\lambda^{-1} \begin{bmatrix} 0 \\ \epsilon \end{bmatrix} = [0 \ \epsilon] \begin{bmatrix} 1 & 0 \\ 0 & 4/\epsilon^2 \end{bmatrix} \begin{bmatrix} 0 \\ \epsilon \end{bmatrix} = 4.$$

This example shows that indeed the performance achieved by $\mathcal{XY}$ may be similar to the performance of $G$-optimal. Let us now consider a different setting where the two arms $x_1 = [1 \ 0]$ and $x_2 = [1 - \epsilon \ 0]$ are aligned on the same axis. In this case, the problem reduces to a 1-dimensional problem and both strategies would concentrate their allocation on $x_1 = [1 \ 0]$ since it is the arm with larger norm and it may provide a better estimate of $\theta^*$. As a result, while the $G$-allocation has a performance of 1, the $\mathcal{XY}$-allocation over the direction $[\epsilon \ 0]$ has a performance $\epsilon^2$, which can be arbitrarily smaller than 1.

## B Proofs

### B.1 Lemmas

*Proof of Lemma 1.* The proof follows from the fact that if $\mathcal{S}^*(\mathbf{x}_n) \subseteq \mathcal{C}(x^*)$ and $\hat{\theta}_n \in \mathcal{S}^*(\mathbf{x}_n)$ with high probability, then $\hat{\theta}_n \in \mathcal{C}(x^*)$ which implies that $\Pi(\hat{\theta}_n) = x^*$ by definition of the cone $\mathcal{C}(x^*)$. $\qquad\square$

Before proceeding to the proof of Lemma 2 we introduce the following technical tool.

**Proposition 3** (Equivalence-Theorem in [13]). *Define $f(x; \xi) = x^\top M(\xi)^{-1} x$, where $M(\xi)$ is a $d \times d$ non-singular matrix and $x$ is a column vector in $\mathbb{R}^d$. We consider two extremum problems.*

*The first is to choose $\xi$ so that*

$$(1) \quad \xi \quad \text{maximizes} \quad \det M(\xi) \qquad\qquad \text{(D-optimal design)}$$

*The second one is to choose $\xi$ so that*

$$(2) \quad \xi \quad \text{minimizes} \quad \max f(x; \xi) \qquad\qquad \text{(G-optimal design)}$$

*We note that the integral with respect to $\xi$ of $f(x; \xi)$ is $d$; hence, $\max f(x; \xi) \geq d$, and thus a sufficient condition for $\xi$ to satisfy (2) is*

$$(3) \quad \max f(x; \xi) = d.$$

*Statements (1), (2) and (3) are equivalent.*

*Proof of Lemma 2.* **Upper-bound.** We have the following sequence of inequalities

$$\max_{y \in \mathcal{Y}*} \frac{||y||_{\Lambda_\lambda^{-1}}^2}{\Delta^2(y)} \leq \frac{1}{\Delta_{\min}^2} \max_{y \in \mathcal{Y}*} ||y||_{\Lambda_\lambda^{-1}}^2 \leq \frac{4}{\Delta_{\min}^2} \max_{x \in \mathcal{X}} ||x||_{\Lambda_\lambda^{-1}}^2,$$

where the second inequality comes from a triangle inequality on $||y||^2_{\Lambda^{-1}_\lambda}$. Thus we obtain

$$\rho^*(\lambda^*) = \min_{\lambda \in \mathcal{D}^k} \max_{y \in \mathcal{Y}*} \frac{||y||^2_{\Lambda^{-1}_\lambda}}{\Delta^2(y)} \le \frac{4}{\Delta^2_{\min}} \min_{\lambda \in \mathcal{D}^k} \max_{x \in \mathcal{X}} ||x||^2_{\Lambda^{-1}_\lambda} = \frac{4d}{\Delta^2_{\min}},$$

where the last equality follows from the Kiefer-Wolfowitz equivalence theorem presented in Prop. 3.

**Lower-bound.**

We focus on the numerator $y^\top \Lambda^{-1}_\lambda y$. Since $\Lambda_\lambda$ is a positive definite matrix, we define its decomposition $\Lambda_\lambda = Q\Gamma Q^\top$, where $Q$ is an orthogonal matrix and $\Gamma$ is the diagonal matrix containing the eigenvalues. As a result the numerator can be written as

$$y^\top \Lambda^{-1}_\lambda y = y^\top Q\Gamma^{-1} Q^\top y = w^\top \Gamma^{-1} w,$$

where we renamed $Q^\top y = w$. If we denote by $\gamma_{\max}$ the largest eigenvalue of $\Lambda_\lambda$ (i.e., the largest value in $\Gamma$), then

$$w^\top \Gamma^{-1} w \ge 1/\gamma_{\max} w^\top w = 1/\gamma_{\max} ||y||^2.$$

The largest eigenvalue $\gamma_{\max}$ is upper-bounded by the sum of the largest eigenvalues of the matrices $\lambda(x)xx^\top$ which is $\lambda(x)||x||_2$. As a result, we obtain the bound $\gamma_{\max} \le \sum_x \lambda(x)||x||_2 \le L$, since $||x||_2 \le L$ and $\lambda$ is in the simplex. Thus we have

$$\min_{\lambda \in \mathcal{D}^k} \max_{y \in \mathcal{Y}*} \frac{||y||^2_{\Lambda^{-1}_\lambda}}{\Delta^2(y)} \ge \frac{1}{L} \max_{y \in \mathcal{Y}*} \frac{||y||^2}{\Delta(y)^2} \ge \frac{\max_{y \in \mathcal{Y}*} ||y||^2}{L\Delta^2_{\min}}.$$

**Comparison with the K-armed bandit complexity.**

Finally, we show how the sample complexity reduces to the known quantity in the MAB case. If the arms in $\mathcal{X}$ coincide with the canonical basis of $\mathbb{R}^d$, then for any allocation $\lambda$ the design matrix $\Lambda_\lambda$ becomes a diagonal matrix of the form $\mathrm{diag}(\lambda(x_1), \ldots, \lambda(x_K))$. As a result, we obtain

$$H_{\mathrm{LB}} = \min_{\lambda \in \mathcal{D}^k} \max_{y \in \mathcal{Y}*} \frac{||y||^2_{\Lambda^{-1}_\lambda}}{\Delta^2(y)} = \min_{\lambda \in \mathcal{D}^k} \max_{x \in \mathcal{X}-\{x^*\}} \frac{1/\lambda(x) + 1/\lambda(x^*)}{\Delta^2(x)}.$$

If we use the allocation $\lambda(x) = 1/(\nu\Delta^2(x))$ and $\lambda(x^*) = 1/(\nu\Delta_{\min})$, with $\nu = 1/\Delta^2_{\min} + \sum_{x \ne x^*} 1/\Delta^2(x)$, we obtain

$$H_{\mathrm{LB}} \le \max_{x \in \mathcal{X}-\{x^*\}} \frac{\nu\Delta^2(x) + \nu\Delta^2_{\min}}{\Delta^2(x)} = \max_{x \in \mathcal{X}-\{x^*\}} \nu + \nu\frac{\Delta^2_{\min}}{\Delta^2(x)}$$

$$= 2\nu = 2\Big(\frac{1}{\Delta^2_{\min}} + \sum_{x \ne x^*} \frac{1}{\Delta^2(x)}\Big) = 2H_{\mathrm{MAB}}.$$

On the other hand, letting $\tilde{x}$ be the second best arm and $\Delta(x^*) = \Delta_{\min}$, we have that

$$H_{\mathrm{LB}} = \min_{\lambda \in \mathcal{D}^k} \max_{x \ne x*} \frac{1/\lambda(x) + 1/\lambda(x^*)}{\Delta^2(x)}$$

$$= \min_{\lambda \in \mathcal{D}^k} \max \left\{ \max_{x \ne x*} \frac{1/\lambda(x) + 1/\lambda(x^*)}{\Delta^2(x)}; \frac{1/\lambda(\tilde{x}) + 1/\lambda(x^*)}{\Delta^2(x^*)} \right\}$$

$$\ge \min_{\lambda \in \mathcal{D}^k} \max \left\{ \max_{x \ne x*} \frac{1/\lambda(x)}{\Delta^2(x)}; \frac{1/\lambda(x^*)}{\Delta^2(x^*)} \right\}$$

$$= \min_{\lambda \in \mathcal{D}^k} \max_{x \in \mathcal{X}} \frac{1/\lambda(x)}{\Delta^2(x)}.$$

We set $\frac{1/\lambda(x)}{\Delta^2(x)}$ equal to a constant $c$ and thus we get $\lambda(x) = \frac{1}{c\Delta^2(x)}$. Since $\frac{1}{c} \sum_{x \in \mathcal{X}} \frac{1}{\Delta^2(x)} = 1$, it follows that:

$$c = \sum_{x \in \mathcal{X}} \frac{1}{\Delta^2(x)} = \sum_{x \ne x^*} \frac{1}{\Delta^2(x)} + \frac{1}{\Delta^2_{\min}} = H_{\mathrm{MAB}}.$$

Thus, we get that $H_{\mathrm{MAB}} \leq H_{\mathrm{LB}} \leq 2H_{\mathrm{MAB}}$. This shows that $H_{\mathrm{LB}}$ is a well defined notion of complexity for the linear best-arm identification problem and the corresponding sample complexity $N^*$ is coherent with existing results in the MAB case. $\qquad\square$

*Proof of Lemma 3.* The proof follows from the fact that if $\widehat{\mathcal{S}}(\mathbf{x}_n) \subseteq \mathcal{C}(x)$ and $\theta^* \in \widehat{\mathcal{S}}(\mathbf{x}_n)$ with high probability, then $\theta^* \in \mathcal{C}(x)$ which implies that $\Pi(\hat{\theta}_n) = x = x^*$. $\qquad\square$

## B.2 Proofs of Theorem 1 and Theorem 2

*Proof of Theorem 1.* The statement follows from Prop. 1 and the performance guarantees for the different implementations of the $G$-optimal design. By recalling the empirical stopping condition in Eq. 11 and the definition $\rho^G(\lambda) = \max_x x^\top \Lambda_\lambda^{-1} x$, we notice that from a simple triangle inequality applied to $||y||_{A^{-1}}$, a sufficient condition for stopping is that for any $x \in \mathcal{X}$

$$\frac{4c^2 \rho_n^{\tilde{G}} \log_n(K^2/\delta)}{n} \leq \widehat{\Delta}_n^2(x^*, x),$$

where $\rho_n^{\tilde{G}} = \rho^G(\lambda_{\mathbf{x}_n^{\tilde{G}}})$ and $\mathbf{x}_n^{\tilde{G}}$ is the allocation obtained from rounding the optimal design $\lambda^G$ obtained from the continuous relaxation or the greedy incremental algorithm. From Prop. 1 we have that the following inequalities

$$\widehat{\Delta}_n(x^*, x) \geq \Delta(x^*, x) - c||x^* - x||_{A_{\mathbf{x}_n^G}^{-1}} \sqrt{\log_n(K^2/\delta)} \geq \Delta(x^*, x) - 2c\sqrt{\frac{\rho_n^{\tilde{G}} \log_n(K^2/\delta)}{n}},$$

hold with probability $1 - \delta$. Combining this with the previous condition and since the condition must hold for all $x \in \mathcal{X}$, we have that a sufficient condition to stop using the $G$-allocation is

$$\frac{16c^2 \rho_n^{\tilde{G}} \log_n(K^2/\delta)}{n} \leq \Delta_{\min},$$

which defines the level of accuracy that the $G$-allocation needs to achieve before stopping. Since $\rho_n^{\tilde{G}} \leq (1 + \beta)d$ then the statement follows by inverting the previous inequality. $\qquad\square$

*Proof of Theorem 2.* We follow the same steps as in the proof of Theorem 1. $\qquad\square$

## C  Implementation of the Allocation Strategies

In this section we discuss about possible implementations of the allocation strategies illustrated in sections 4 and 5 and we discuss their approximation accuracy guarantees.

**The efficient rounding procedure.** We first report the general structure of the efficient rounding procedure defined in [15, Chapter 12] to implement a design $\lambda$ into an allocation $\mathbf{x}_n$ for any fixed number of steps $n$. Let $p = \mathrm{supp}(\lambda)$ the support of $\lambda$,[4] then we want to compute the number of pulls $n_i$ (with $i = 1, \ldots, p$) for all the arms in the support of $\lambda$. Basically, the fast implementation of the design is obtained in two phases, as follows:

- In the first phase, given the sample size $n$ and the number of support points $p$, we calculate their corresponding frequencies $n_i = \lceil (n - \frac{1}{2}p)\lambda_i \rceil$, where $n_1, n_2, \ldots, n_p$ are positive integers with $\sum_{i \leq p} n_i \geq n$.
- The second phase loops until the discrepancy $\left( \sum_{i \leq p} n_i \right) - n$ is 0, either:
  - increasing a frequency $n_j$ which attains $n_j/\lambda_j = \min_{i \leq p}(n - 1)/\lambda_i$ to $n_{j+1}$, or
  - decreasing some $n_k$ with $(n_k - 1)/\lambda_k = \max_{i \leq p}(n_i - 1)/\lambda_i$ to $n - 1$.

An interesting feature of this procedure is that when moving from $n$ to $n + 1$ the corresponding allocations $\mathbf{x}_n$ and $\mathbf{x}_{n+1}$ only differ for one element $i$ which is increased by 1, i.e., the discrete allocation is monotonic in $n$.

**Implementation of the G-allocation.** A first option is to optimize a continuous relaxation of the problem and compute the optimal design. Let $\rho^G(\lambda) = \max_x x^\top \Lambda_\lambda^{-1} x$, the optimal design is

$$\lambda^G = \arg \min_{\lambda \in \mathcal{D}_k} \max_{x \in \mathcal{X}} ||x||^2_{\Lambda_\lambda^{-1}} = \arg \min_{\lambda \in \mathcal{D}_k} \rho^G(\lambda). \tag{21}$$

This is a convex optimization problem and it can be solved using the projected gradient algorithm, interior point techniques, or multiplicative algorithms. To move from the design $\lambda^G$ to a discrete allocation we use the efficient rounding technique presented above and we obtain that the resulting allocation $\mathbf{x}_t^{\tilde{G}}$ is guaranteed to be monotonic as the number of times an arm $x$ is pulled is non-decreasing with $t$. Thus from $\mathbf{x}_t^{\tilde{G}}$ we obtain a simple incremental rule, where the arm $x_t$ is the arm for which $\mathbf{x}_t^{\tilde{G}}$ recommends one pull more than in $\mathbf{x}_{t-1}^{\tilde{G}}$. An alternative is to directly implement an incremental version of Eq. 12 by selecting at each step $t$ the greedy arm

$$x_t = \arg \min_{x \in \mathcal{X}} \max_{x' \in \mathcal{X}} x'^\top (A_{\mathbf{x}_{t-1}} + xx^\top)^{-1} x' = \arg \min_{x \in \mathcal{X}} \max_{x' \in \mathcal{X}} x'^\top \left[ A_{\mathbf{x}_{t-1}}^{-1} - \frac{A_{\mathbf{x}_{t-1}}^{-1} xx^\top A_{\mathbf{x}_{t-1}}^{-1}}{1 + x^\top A_{\mathbf{x}_{t-1}}^{-1} x} \right] x', \tag{22}$$

where the second formulation follows from the matrix inversion lemma. This allocation is somehow simpler and more direct than using the continuous relaxation but it may come with a higher efficiency loss.

Before reporting the performance guarantees for the two implementations proposed above, we introduce an additional technical lemma which will be useful in the proofs on the performance guarantees. Although the lemma is presented for a specific definition of uncertainty $\rho$, any other notion including design matrices of the kind $\Lambda_\lambda$ will satisfy the same guarantee.

**Lemma 5.** *Let $\rho(\lambda) = \max_{x \in \mathcal{X}} x^\top \Lambda_\lambda^{-1} x$ be a measure of uncertainty of interest for any design $\lambda \in \mathcal{D}^K$. We denote by $\lambda^* = \arg \min_{\lambda \in \mathcal{D}^K} \rho(\lambda)$ the optimal design and for any $n > d$ we introduce the optimal discrete allocation as*

$$\mathbf{x}_n^* = \arg \min_{\mathbf{x}_n \in \mathcal{X}^n} \max_{x \in \mathcal{X}} \frac{x^\top \Lambda_{\lambda_{\mathbf{x}_n}}^{-1} x}{n},$$

*where $\lambda_{\mathbf{x}_n}$ is the (fractional) design corresponding to $\mathbf{x}_n$. Then we have*

$$\rho(\lambda^*) \le \rho(\mathbf{x}_n^*) \le \left(1 + \frac{p}{n}\right)\rho(\lambda^*), \tag{23}$$

*where $p = supp(\lambda^*)$ is the number of points in the support of $\lambda^*$. If $d$ linearly independent arms are available in $\mathcal{X}$, then we can upper bound the size of the support of $\lambda^*$ and obtain*

$$\rho(\lambda^*) \le \rho(\mathbf{x}_n^*) \le \left(1 + \frac{d(d+1)}{n}\right)\rho(\lambda^*). \tag{24}$$

*Proof.* The first part of the statement follows by the definition of $\lambda^*$ as the minimizer of $\rho$. Let $\tilde{\mathbf{x}}_n$ by an efficient rounding technique applied on $\lambda^*$ such as the one described in Lemma 12.8 in [15]. Then $\tilde{\mathbf{x}}_n$ has the same support as $\lambda^*$ and an efficiency loss bounded by $p/n$. As a result, we have

$$\rho(\mathbf{x}_n^*) \le \rho(\tilde{\mathbf{x}}_n) \le \left(1 + \frac{p}{n}\right)\rho(\lambda^*),$$

where the first inequality comes from the fact that $\mathbf{x}_n^*$ is the minimizer of $\rho$ among allocations of length $n$. Then, from Caratheodory's theorem (see e.g., [15] ) the number of support points used in $\lambda^*$ is upper bounded by $p \le d(d+1)/2 + 1$ (under the assumption that there are $d$ linearly independent arms in $\mathcal{X}$). The final result follows by a rough maximization of $d(d+1)/2n + 1/n \le d(d+1)/n$. $\square$

**Remark 1.** Note that the same upper-bound for the number of support points holds for any design, due to the properties of the design matrices. In fact, any design matrix is symmetric by construction, which implies that it is completely described by $D = d(d+1)/2$ elements and can thus be seen as a point in $\mathbb{R}^D$. Moreover, a design matrix is a convex combination of a subset of points in $\mathbb{R}^D$ and thus it belongs to the convex hull of that subset of points. Caratheodory's theorem states that each point in the convex hull of any subset of points in $\mathbb{R}^D$ can be defined as a convex combination of at

most $D+1$ points. It directly follows that any design matrix can be expressed using $(d(d+1)/2)+1$ points.

It follows that the allocation $\mathbf{x}_t^{\tilde{G}}$ obtained applying the rounding procedure has the following performance guarantee.

**Lemma 6.** *For any $t \geq d$, the rounding procedure defined in [15, Chapter 12] returns an allocation $\mathbf{x}_t^{\tilde{G}}$, whose corresponding design $\lambda^{\tilde{G}} = \lambda_{\mathbf{x}_t^{\tilde{G}}}$ is such that[5]*

$$\rho^G(\lambda^{\tilde{G}}) \leq \left(1 + \frac{d + d^2 + 2}{2t}\right)d.$$

*Proof of Lemma 6.* We follow the same steps as in the proof of Lemma 5 to obtain the term $\beta = \frac{d+d^2+2}{2t}$. Then, noting that the performance of the optimal strategy $\rho^G(\lambda^{*G}) = d$ (from Prop. 3), the results follows. $\square$

**Implementation of the $\mathcal{XY}$-allocation.** Notice that the complexity of the $\mathcal{XY}$-allocation trivially follows from the complexity of the $G$-allocation and it is NP-hard. As a result, we need to propose approximate solutions to compute an allocation $\mathbf{x}_n^{\widetilde{\mathcal{XY}}}$ as for the $G$-allocation. Let $\rho^{\mathcal{XY}}(\lambda) = \max_{y \in \mathcal{Y}} y^\top \Lambda_\lambda^{-1} y$, then the first option is the compute the optimal solution to the continuous relaxed problem

$$\lambda^{\mathcal{XY}} = \arg\min_{\lambda \in \mathcal{D}_k} \max_{y \in \mathcal{Y}} ||y||_{\Lambda_\lambda^{-1}}^2 = \arg\min_{\lambda \in \mathcal{D}_k} \rho^{\mathcal{XY}}(\lambda). \tag{25}$$

And then compute the corresponding discrete allocation $\mathbf{x}_n^{\widetilde{\mathcal{XY}}}$ using the efficient rounding procedure. Alternatively, we can use an incremental greedy algorithm which at each step $t$ returns the arm

$$x_t = \arg\min_{x \in \mathcal{X}} \max_{y \in \mathcal{Y}} y^\top \left(A_{\mathbf{x}_{t-1}} + xx^\top\right)^{-1} y. \tag{26}$$

**Lemma 7.** *For any $t \geq d$, the rounding procedure defined in [15, Chapter 12] returns an allocation $\mathbf{x}_t^{\widetilde{\mathcal{XY}}}$, whose corresponding design $\lambda^{\widetilde{\mathcal{XY}}} = \lambda_{\mathbf{x}_t^{\widetilde{\mathcal{XY}}}}$ is such that*

$$\rho^{\mathcal{XY}}(\lambda^{\widetilde{\mathcal{XY}}}) \leq 2\left(1 + \frac{d + d^2 + 2}{2t}\right)d.$$

*Proof of Lemma 7.* The proof follows from the fact that for any pair $(x, x')$

$$||x - x'||_{A_{\mathbf{x}_n}^{-1}} \leq 2 \max_{x'' \in \mathcal{X}} ||x''||_{A_{\mathbf{x}_n}^{-1}}.$$

Then the proof proceeds as in Lemma 6. $\square$

**Implementation of $\mathcal{XY}$-adaptive allocation.** The allocation rule in Eq. 17 basically coincides with the $\mathcal{XY}$-allocation and its properties extend smoothly.

# D  Proof of Theorem 3

Before proceeding to the proof, we first report the proofs of two aditional lemmas.

*Proof of Lemma 4.* Let $y = x' - x$. Using the definition of $\widehat{\mathcal{S}}(\mathbf{x}_n)$ in Eq. 10, and the fact that $\theta^* \in \widehat{\mathcal{S}}(\mathbf{x}_n)$ with high probability, we have

$$(x' - x)^\top (\hat{\theta}_n - \theta^*) \leq c||x' - x||_{A_{\mathbf{x}}^{-1}} \sqrt{\log_n(K^2/\delta)}.$$

Since the condition in Eq. 16 is true, it follows that

$$(x' - x)^\top (\hat{\theta}_n - \theta^*) \leq c||x' - x||_{A_{\mathbf{x}}^{-1}} \sqrt{\log_n(K^2/\delta)} \leq \widehat{\Delta}_n(x', x) \Leftrightarrow$$
$$-(x' - x)^\top \theta^* \leq 0 \Leftrightarrow x^\top \theta^* \leq x'^\top \theta^*$$

thus $x$ is dominated by $x'$ and $x$ cannot be the optimal arm. $\square$

**Lemma 8.** *For any phase $j$, the length is such that $n_j \leq \max\{M^*, \frac{16}{\alpha} N^*\}$ with probability $1 - \delta$.*

*Proof of Lemma 8.* We first summarize the different quantities measuring the performance of an allocation strategy in different settings. For any design $\lambda \in \mathcal{D}^K$, we define

$$\rho^*(\lambda) = \max_{y \in \mathcal{Y}^*} \frac{||y||^2_{\Lambda_\lambda^{-1}}}{\Delta^2(y)}; \quad \rho^{\mathcal{X}\mathcal{Y}}(\lambda) = \max_{y \in \mathcal{Y}} ||y||^2_{\Lambda_\lambda^{-1}}; \quad \rho^j(\lambda) = \max_{y \in \widehat{\mathcal{Y}}_j} ||y||^2_{\Lambda_\lambda^{-1}}. \tag{27}$$

For any $n$, we also introduce the value of each of the previous quantities when the corresponding optimal (discrete) allocation is used

$$\rho_n^* = \rho^*(\lambda_{\mathbf{x}_n^*}); \quad \rho_n^{\mathcal{X}\mathcal{Y}} = \rho^{\mathcal{X}\mathcal{Y}}(\lambda_{\mathbf{x}_n^{\mathcal{X}\mathcal{Y}}}); \quad \rho_n^j = \rho^j(\lambda_{\mathbf{x}_n^j}). \tag{28}$$

Finally, we introduce the optimal designs

$$\lambda^* = \arg\min_{\lambda \in \mathcal{D}^K} \rho^*(\lambda); \quad \lambda^{\mathcal{X}\mathcal{Y}} = \arg\min_{\lambda \in \mathcal{D}^K} \rho^{\mathcal{X}\mathcal{Y}}(\lambda); \quad \lambda^j = \arg\min_{\lambda \in \mathcal{D}^K} \rho^j(\lambda). \tag{29}$$

Let $\epsilon^*$ be the smallest $\epsilon$ such that there exists a pair $(x, x')$, with $x \neq x^*$ and $x' \neq x^*$, such that the confidence set $\mathcal{S} = \{\theta : \forall y \in \mathcal{Y}, |y^\top(\theta - \theta^*)| \leq \epsilon\}$ overlaps with the hyperplane $\mathcal{C}(x) \cap \mathcal{C}(x')$. Since $M^*$ is defined as the smallest number of steps needed by the $\mathcal{X}\mathcal{Y}$ strategy to avoid any overlap between $\mathcal{S}^*$ and the hyperplanes $\mathcal{C}(x) \cap \mathcal{C}(x')$, then we have that after $M^*$ steps

$$c\sqrt{\frac{\rho_{M^*}^{\mathcal{X}\mathcal{Y}} \log_n(K^2/\delta)}{M^*}} < \epsilon^*. \tag{30}$$

We consider two cases to study the length of a phase $j$.

**Case 1:** $\sqrt{\frac{\rho_{n_j}^j}{n_j}} \geq \frac{\epsilon^*}{c\sqrt{\log_n(K^2/\delta)}}$. From Eq. 30 it immediately follows that

$$\frac{\rho_{n_j}^j}{n_j} \geq \frac{\rho_{M^*}^{\mathcal{X}\mathcal{Y}}}{M^*}. \tag{31}$$

From definitions in Eqs. 27 and 28, since $\widehat{\mathcal{Y}}_j \subseteq \mathcal{Y}$ we have for any $n$, $\rho_n^j \leq \rho_n^{\mathcal{X}\mathcal{Y}}$. As a result, if $n_j \geq M^*$, since $\rho_n^j/n$ is a non-increasing function, then we would have the sequence of inequalities

$$\frac{\rho_{n_j}^j}{n_j} \leq \frac{\rho_{M^*}^j}{M^*} \leq \frac{\rho_{M^*}^{\mathcal{X}\mathcal{Y}}}{M^*},$$

which contradicts Eq. 31. Thus $n_j \leq M^*$.

**Case 2:** $\sqrt{\frac{\rho_{n_j}^j}{n_j}} \leq \frac{\epsilon^*}{c\sqrt{\log_n(K^2/\delta)}}$. We first relate the performance at phase $j$ with the performance of the oracle. For any $n$

$$\rho_n^j = \rho^j(\lambda_{\mathbf{x}_n^j}) \leq \rho^j(\lambda_{\mathbf{x}_n^*}) = \max_{y \in \widehat{\mathcal{Y}}_j} y^\top \Lambda_{\lambda_{\mathbf{x}_n^*}}^{-1} y = \max_{y \in \widehat{\mathcal{Y}}_j} \frac{y^\top \Lambda_{\lambda_{\mathbf{x}_n^*}}^{-1} y}{\Delta^2(y)} \Delta(y) \leq \max_{y \in \widehat{\mathcal{Y}}_j} \frac{y^\top \Lambda_{\lambda_{\mathbf{x}_n^*}}^{-1} y}{\Delta^2(y)} \max_{y \in \widehat{\mathcal{Y}}_j} \Delta^2(y).$$

If now we consider $n = n_j$, then the definition case 2 implies that the estimation error $\sqrt{\rho_{n_j}^j/n_j}$ is small enough so that all the directions in $\mathcal{Y} - \mathcal{Y}^*$ have already been discarded from $\widehat{\mathcal{Y}}_j$ and $\widehat{\mathcal{Y}}_j \subseteq \mathcal{Y}^*$. Thus

$$\rho_{n_j}^j \leq \max_{y \in \mathcal{Y}^*} \frac{y^\top \Lambda_{\lambda_{\mathbf{x}_{n_j}^*}}^{-1} y}{\Delta^2(y)} \max_{y \in \widehat{\mathcal{Y}}_j} \Delta^2(y) = \rho_{n_j}^* \max_{y \in \widehat{\mathcal{Y}}_j} \Delta^2(y). \tag{32}$$

This relationship does not provide a bound on $n_j$ yet. We first need to recall from Prop. 1 that for any $y \in \mathcal{Y}$ (and notably for the directions in $\widehat{\mathcal{Y}}_j$) we have

$$|y^\top(\hat{\theta}_{j-1} - \theta^*)| \leq c\sqrt{y^\top A_{j-1}^{-1} y \log_n(K^2/\delta)},$$

where $A_{j-1} = A_{\mathbf{x}_{n_{j-1}}^{j-1}}$ is the matrix constructed from the pulls within phase $j-1$. Since $\mathbf{x}_n^{j-1}$ is obtained from a $\mathcal{XY}$-allocation applied on directions in $\widehat{\mathcal{Y}}_{j-1}$, we obtain that for any $y \in \widehat{\mathcal{Y}}_j$

$$|y^\top(\hat{\theta}_{j-1} - \theta^*)| \leq c\sqrt{\log_n(K^2/\delta)} \max_{y \in \widehat{\mathcal{Y}}_{j-1}} \sqrt{y^\top A_{j-1}^{-1} y} = c\sqrt{\frac{\log_n(K^2/\delta)\rho_{n_{j-1}}^{j-1}}{n_{j-1}}},$$

Reordering the terms in the previous expression we have that for any $y \in \widehat{\mathcal{Y}}_j$

$$\Delta(y) \leq \widehat{\Delta}_{j-1}(y) + c\sqrt{\frac{\log_n(K^2/\delta)\rho_{n_{j-1}}^{j-1}}{n_{j-1}}}.$$

Since the direction $y$ is included in $\widehat{\mathcal{Y}}_j$ then the discard condition in Eq. 16 failed for $y$, implying that $\widehat{\Delta}_{j-1}(y) \leq c\sqrt{\frac{\log_n(K^2/\delta)\rho_{n_{j-1}}^{j-1}}{n_{j-1}}}$. Thus we finally obtain

$$\max_{y \in \widehat{\mathcal{Y}}_j} \Delta(y) \leq 2c\sqrt{\frac{\log_n(K^2/\delta)\rho_{n_{j-1}}^{j-1}}{n_{j-1}}}.$$

Combining this with Eq. 32 we have

$$\rho_{n_j}^j \leq \rho_{n_j}^* 4c^2 \frac{\log_n(K^2/\delta)\rho_{n_{j-1}}^{j-1}}{n_{j-1}}.$$

Using the stopping condition of phase $j$ and the relationship between the performance $\rho^j$, we obtain that at time $\bar{n} = n_j - 1$

$$\frac{\rho_{\bar{n}}^j}{\bar{n}} \geq \alpha\frac{\rho_{n_{j-1}}^{j-1}}{n_{j-1}} \geq \frac{\alpha}{4c^2 \log_n(K^2/\delta)} \frac{\rho_{n_j}^j}{\rho_{n_j}^*}$$

We can further refine the previous inequality as

$$\frac{\rho_{\bar{n}}^j}{\bar{n}} \geq \frac{\alpha\rho_{N^*}^*}{4N^*} \frac{N^*}{c^2 \log_n(K^2/\delta)\rho_{N^*}^*} \frac{\rho_{n_j}^j}{\rho_{n_j}^*} \geq \frac{\alpha\rho_{N^*}^*}{4N^*} \frac{\rho_{n_j}^j}{\rho_{n_j}^*},$$

where we use the definition of $N^*$ in Eq. 7, which implies $c\sqrt{\log_n(K^2/\delta)\rho_{N^*}^*/N^*} \leq 1$. Reordering the terms and using $\bar{n} = n_j - 1$, we obtain

$$n_j \leq 1 + \frac{4N^*}{\alpha} \frac{\rho_{n_j-1}^j}{\rho_{n_j}^j} \frac{\rho_{n_j}^*}{\rho_{N^*}^*}.$$

From Lemma 5 and the optimal designs defined in Eq. 29 we have

$$n_j \leq 1 + \frac{4N^*}{\alpha} \frac{(1 + d(d+1)/(n_j-1))\rho^j(\lambda^j)}{\rho^j(\lambda^j)} \frac{(1 + d(d+1)/(n_j-1))\rho^*(\lambda^*)}{\rho^*(\lambda^*)}.$$

Using the fact that the algorithm forces $n_j \geq d(d+1) + 1$, the statement follows. $\qquad\square$

*Proof of Theorem 3.* Let $J$ be the index of any phase for which $|\widehat{\mathcal{X}}_J| > 1$. Then there exist at least one arm $x \in \mathcal{X}$ (beside $x^*$) for which the discarding condition in Lemma 4 is not triggered, which corresponds to the fact that for all arms $x' \in \mathcal{X}$

$$c\|x - x'\|_{A_{\mathbf{x}_{n_J}^J}^{-1}} \sqrt{\log_n(K^2/\delta)} \geq \widehat{\Delta}_J(x, x').$$

By developing the right hand side, we have

$$\widehat{\Delta}_J(x, x') \geq \Delta(x, x') - c\|x - x'\|_{A_{\mathbf{x}_{n_J}^J}^{-1}} \sqrt{\log_n(K^2/\delta)} \geq \Delta_{\min} - c\sqrt{\frac{\rho_{n_J}^J \log_n(K^2/\delta)}{n_J}}$$

which leads to the condition

$$2c\sqrt{\frac{\rho_{n_J}^J \log_n(K^2/\delta)}{n_J}} \geq \Delta_{\min}. \tag{33}$$

Using the phase stopping condition and the initial value of $\rho^0$ we have

$$\frac{\rho_{n_J}^J}{n_J} \leq \alpha \frac{\rho_{n_{J-1}}^{J-1}}{n_{J-1}} \leq \alpha^J \frac{\rho^0}{n_0} = \alpha^J.$$

By joining this inequality with Eq. 33 we obtain

$$\alpha^J \geq \frac{\Delta_{\min}^2}{4c^2 \log_n(K^2/\delta)},$$

and it follows that $J \leq \log(4c^2 \log_n(K^2/\delta)/\Delta_{\min}^2)/\log(1/\alpha)$ which together with Lemma 8 leads to the final statement. $\qquad\square$

## E   Additional Empirical Results

For the setting described in Sec. 6, in order to point out the different repartitions of the sampling budget over arms, in Fig. 5 we present the number of samples allocated per arm, for the case when the input space $\mathcal{X} \subseteq \mathbb{R}^5$. We remind that the arms denoted $x_1, \ldots, x_5$ form the canonical basis and arm $x_6 = [\cos(\omega) \quad \sin(\omega) \quad 0 \quad 0 \quad 0]$.

| Samples/arm | $\mathcal{XY}$-oracle | $\mathcal{XY}$-adaptive | $\mathcal{XY}$ | $G$ | Fully-adaptive |
|---|---|---|---|---|---|
| $x_1$ | 207 | 263 | 29523 | 28014 | 740 |
| $x_2$ | 41440 | 52713 | 29524 | 28015 | 149220 |
| $x_3$ | 2 | 3 | 29524 | 28015 | 1 |
| $x_4$ | 2 | 5 | 29524 | 28015 | 1 |
| $x_5$ | 1 | 2 | 29524 | 28015 | 1 |
| $x_6$ | 0 | 2 | 1 | 1 | 1 |
| **Budget** | 41652 | 52988 | 147620 | 140075 | 149964 |

Figure 5: The budget needed by the allocation strategies to identify the best arm when $\mathcal{X} \subseteq \mathbb{R}^5$ and their sample allocation over arms. $\mathcal{XY}$ and $G$ allocate samples uniformly over the canonical arms while $\mathcal{XY}$-oracle and $\mathcal{XY}$-adaptive use most of the samples for arm $x_2$ (corresponding to the most informative direction).

We can notice that even though the Fully-adaptive algorithm identifies the most informative direction and focuses the sampling on arm $x_2$, its sample complexity still has a growth linear in the dimension, due to the extra $\sqrt{d}$ term in his bound. Consequently, the advantage over the static strategies is canceled. On the other hand, $\mathcal{XY}$-adaptive "learns" the gaps from the observations and allocates the samples very similarly to $\mathcal{XY}$-oracle, without suffering a large loss in terms of the sampling budget. However, $\mathcal{XY}$-adaptive's sample complexity has to account for the the re-initializations made at the beginning of a new phase.

Finally, we notice that in this problem that static allocations, $\mathcal{XY}$ and $G$, perform a uniform allocation over the canonical arms. Another interesting remark is that the number of pulls to one canonical arm is smaller than the samples that $\mathcal{XY}$-oracle allocated to $x_2$. This is explained by the "mutual information" coming from the multiple observations on all directions, which helps in reducing the overall uncertainty of the confidence set.

## Footnotes

[4]For a fixed design $\lambda \in \mathbb{R}^K$, we say that its *support* is given by all arms in $\mathcal{X}$ whose corresponding features in $\lambda$ are different than 0.

[5]We recall that from any allocation $\mathbf{x}_n$ the corresponding design $\lambda_{\mathbf{x}}$ is such that $\lambda_{\mathbf{x}_n}(x) = T_n(x)/n$.