[Reviews · NeurIPS 2014]

Submitted by Assigned_Reviewer_4

Problem definition:- The paper studies best-arm identification in linear bandits. In this problem there is a hidden unknown parameter \theta*\in R^d and a finite set of arms X\subseteq R^d. When an arm is pulled you observe a reward x^T\theta*+epsilon where epsilon is a zero mean i.i.d noise with bounded range. The goal is to identify argmax_{x\in X} x^T\theta* with the least number of samples.

Results: The paper characterizes the sample complexity of static and dynamic allocation strategies to identify the best arm.

Quality of the paper:- The techniques appear reasonably sound. In the experiments section it would also be nice to compare to the existing algorithms for linear bandits. (i.e run the existing algorithm for linear bandit till \hat{S}\subseteq C(x) and then count the number of steps to achieve this). The reason this comparison is important is that the paper mentions at the beginning of the paper that unlike MAB, the strategies to minimize sample complexity of best arm identification of LB could be different from regret minimization.

Clarity of the paper:- The paper looks reasonably well written.

Significance:- The paper proposes and solves a reasonably general problem. The paper does not necessarily do a good job mentioning applications, but I think it looks like an abstract problem which should have good applications.
Summary: The paper studies best-arm identification in linear bandits. It characterizes the sample complexity of static and dynamic allocation strategies to identify the best arm.

Submitted by Assigned_Reviewer_7

This paper is one of the first to study best arm identification in bandit models with correlated arms. The authors consider the fixed-confidence setting. A new complexity notion, that takes into account the correlation between arms, is proposed, and algorithms that almost attain this complexity are proposed. Some of them are based on a static allocation strategy (i.e. that does not depend on the rewards collected), and one of them is adaptive.

First, this paper is not exactly the first on best arm identification in linear bandit problem. Indeed,

* Hoffmann, Shahriari, De Freitas, On correlation and budget constraints in model-based bandit optimization with application to automatic machine learning (AISTATS 2014)

consider a linear bandit model with gaussian noise. However, they study the fixed-budget setting, and the complexity term featured in their work involves a sum over all arms of a squared gap (that is, it does not really take into account the correlations induced by the linear structured).

The theoretical results of the paper rely on Propositions 1 and 2 given in Section 2. The statement of these results are a bit imprecise, and it should be mentionned whether n is fixed or not. Indeed, the result extracted from [1] given as Proposition 2 should be

P(\forall n\in\N, (2) holds) \geq 1-\delta,

whereas Proposition 1 should be written as: for every n, for every fixed sequence x_n,

P( (1) holds) \geq 1- \delta.

This makes a difference, as an extra union bound over n might be required when using Proposition 1, as even when the sampling rule is static, the sampling rule is adaptive. Also a precise reference for Proposition 1 should be given.

The terminology should be made more precise. In best arm identification, in the fixed-confidence setting, algorithms consists in two elements: a sampling rule and a stopping rule (and also a recommendation rule, but quite naturally, when stopping you always choose the best arm as if the current least-square estimate were the true parameter). The "static and adaptive strategies" mentioned in the paper relates only to the sampling strategy as the stopping strategy of the G-allocation strategy is adaptive. In the proof of Theorem 1, a union bound over n seems to be missing (see above comment on Proposition 1): you need the display of line 674 to hold for all n, hence the union bound will bring you a log(CK^2/t^2\delta) for some constant C for example. Besides, in the statement of Theorem 1, 'beta-approximation' is not defined (neither in the text nor in Lemma 5 and 6).

About the oracle: Section 3 is a bit confusing as when you have access to the parameter \theta^*, there is nothing left to do. I understand that you assume that the algorithms are based on confidence intervals and stop when one of the confidence intervals is contained on the C(x). You then study which shape these confidence intervals should have in order to be as quickly as possible included in C(x^*). Still the definition of the oracle appears quite imprecise (again if \theta^* was known the problem would be solved). Usually the definition of a complexity in best arm identification comes from a lower bound on the number of sample used, for any $\delta$-PAC algorithm: you don't really have such a result here and I don't see what justifies the claim "We characterize the complexity of the problem" in the introduction.

The algorithms are built on allocation strategies coming from experimental design theory (many of the results of Appendix B are adapted from [16]) which are hard to implement and must be approximated. It is not clear what is the overall numerical complexity of the proposed algorithm and how it scales with the different parameters.

In the numerical experiments section, it is always frustrating to see comparison only between the methods introduced in the paper. As such, this section is not very informative. In particular, an obvious benchmark would have been to compare to an algorithm designed for best arm identification in classic (unstructured) bandit problems (LUCB, UGapE, etc.) This is particularly true as the problem chosen in the experiments is such that the arms are orthogonal, except for two of them; a situation which is very close to the classic d-armed bandit model. Comments about the choice of such a particular setting for the simulations would also be appreciable.

Minor comments

* There is an overlap in the notation: in the paragraph "the setting", \Delta is defined in two ways, \Delta(x) and \Delta(y) (one beeing functions of vectors, the others of directions): it might be more convenient for the reader to distinguish explicitly these two notions.
* In Footnote 1 "used of all direction y": should it be "for all vector x"?
* Line 232, "inthe" (should be separated).
* In the boxed environment of page 4 (Figure 2), it should be Eq. 24 in place of Eq. 22 on the second line.
* Figures 2 and 3 (which actually define the algorithms) refer to Equations that only appear in the supplementary material; please make the main paper self-contained.
Summary: This paper addresses the challenging issue of best-arm identification in linear bandit models. It is novel and uses some insights from the theory of optimal design. Still it does not really provide a generic notion of sample complexity and provides methods whose complexity appears to be high. The experimental results are also very limited.

Submitted by Assigned_Reviewer_41

The paper deals with the pure exploration variant of linear bandits. Here, the input consists of K vectors of d dimensions. The player is allowed to query vectors and obtain a sample from a distribution whose mean is a fixed linear function of the vector. The objective is to find the arm with maximum value w.h.p using a minimal number of queries.

The authors provide a solution to the problem based on the experiment design framework. They offer two static allocation strategies, meaning methods for querying the arms that are not adaptive to the outcome of the queries. These strategies provide a result analogous to that of the uniform strategy w.r.t the classic best arm identification problem. The authors also provide a dynamic strategy achieving a provable result that can outperform the static strategy in some scenarios. In addition to the proposed solution, a problem-dependent optimal strategy is given that matches that of the classic best arm identification problem.

The main body of the paper is in general well written. Some of the proofs in the appendix can use a few more details but they hold, as far as I checked. The studied problem is interesting and the paper provides an extensive theoretical study of it. However, I found the experiments a bit lacking. The result for the adaptive sampling technique is a bit weak in the sense that there is a separation between it and static methods only in a very specific case in which, among other requirements, finding the second best arm is a much easier task than finding the best arm. It is an interesting question whether the true performance of the adaptive setting is indeed superior to the static only in this restricted scenario or whether it is superior in other scenarios as well. Proving such a statement may prove to be difficult but there should be experiments aimed to answer this question. Instead the experiments are made w.r.t an example carefully built to match the conditions in which the adaptive algorithm is provably superior.

Comments / typos:
• Proposition 1: should be “w.p. 1-\delta, for all x \in …” rather than the other way
• Line 101: D^k – small k rather than large K
• Equations 3,4,5: It seems that the expression in the log be K/\delta rather than K^2/\delta.
• Given the optimal strategy, it seems rather easy to prove a lower bound for the sample complexity
• In Lemma 2, H_LB is bounded from above and below. It would be interesting to see for some example data sets how close it is to the upper and lower bound
• Theorem 1: N^G should be defined
• Line 269: The comment after theorem 2 is too vauge. It seems that there is some expression “NG < ExpressionG < the given bound” and “NXY < ExpressionXY < the given bound” that is missing, since the bounds as given are exactly the same
• Line 352: missing space
• Section B: Wouldn’t it be easier to simply sample from the strategy provided by the convex program? Matrix chernoff boundes should help ensure (w.h.p) that the sample converged well
• Proof of lemma 6: Theorem 2.6 of the mentioned paper does not necessarily apply for D-optimal design. Also, Some reference is needed for the claim that G and D optimal design are equivalent, even w.r.t approximations (as the D-optimal problem is only approximated by the greedy strategy)
• Equation 30: should be <, not \leq
• Line 753: should be n_j > M^*

Summary: Overall I recommend accepting the paper since the problem is interesting and the analysis is solid and extensive. However, the quality of the paper can potentially increase if the authors can manage to add more meaningful experiments, either on real data or on more plausible synthetic data.
Author Feedback
Author rebuttal: We thank all reviewers for their thoughtful and useful feedback. Here we address the main issues raised by the reviewers. We will fix minor comments and typos in the final version of the paper.

Rev.4

1A "compare to the existing algorithms for linear bandits (LB)"
Since most existing algorithms for LB are designed for cumulative regret minimization (CRM), such comparison would not be totally fair. Intuitively, CRM algorithms pull near-optimal arms as often as possible to keep the regret small. This behavior prevents them from pulling arms that are more "informative" in discovering the best arm. As a result, the sample complexity of CRM algorithms is higher than best-arm identification algorithms (notably XY-Adaptive). For instance, in our experiment a CRM algorithm would mostly pull arms x_1 and x_{d+1}, while the more informative arm x_2 would almost never be pulled. As a result, many more samples are needed before correctly discriminating between x_1 and x_{d+1}.

Rev.41

1B "separation between the adaptive and static sampling techniques only in a very specific case"
The performance of XY-Adaptive is actually better than static strategies in a much wider range of cases than the one considered in our experiment. Intuitively, as soon as XY-Adaptive discards arms/directions, its sample complexity becomes smaller than for a static allocation that sticks to a fixed set of directions. In fact, XY-Adaptive does actually perform the XY-static strategy at the beginning but then it progressively discards arms/directions and focuses on improving the estimates of the gaps only for the remaining arms. It is intuitive that the fewer the "active" arms/directions, the fewer the samples needed to find the optimal arm. Thus, apart from an additional factor 1/log(1/\alpha) due to the re-initializations in each phase, XY-Adaptive cannot be worse than static strategies and is superior whenever at least one direction can be discarded. In the example in Sect.6 we show that the improvement can be up to a factor of d.

2B "seems rather easy to prove a lower bound for the sample complexity"
Please refer to 3C.

3B Example of the bounds of H_LB
The upper bound is achieved in a d-arm MAB problem (ie orthogonal arms) where all arms have the same gap. In this case H_LB is O(d/Delta_{min}^2). As for the lower bound, an example is provided in the experiment where Delta_{min} is very small, all other arms have large gaps and can be easily discarded, and one arm (ie x_2) is almost aligned with the direction connecting optimal and second-optimal arms. In this case H_LB is O(1/Delta_{min}^2).

Rev.7

1C Reference to Hoffmann et al.
We will include in the final version this useful reference for the related fixed-budget setting.

2C "statements of Propositions 1 and 2"
We agree with the reviewer and we will fix this to hold for random n. As suggested in the review, the resulting bound will have an additional factor log(t^2). Since this change is common to all algorithms, high-level discussions are not affected. Proposition 1 is a direct application of Azuma's inequality.

3C "definition of the oracle"
XY-Oracle provides a first characterization of which problem-dependent quantities play an important role in defining the complexity of the problem, but N^* cannot be claimed to be the "real" complexity until a lower-bound is proved. This remains an open and challenging question (notice that even in the MAB setting this is still a partially open question). Nonetheless, besides the intuition that XY-Oracle is a good proxy for an "ideal" learning algorithm, the fact that H_LB reduces to H_MAB and that the performance of XY-Adaptive can be related to it is an evidence that N* may indeed appear in the lower bound. In fact, N* is related to the smallest number of samples needed to fit the confidence set into the optimal cone and we cannot expect any learning algorithm to be able to achieve this any faster.

4C "overall numerical complexity" and "beta-approximation"
For the greedy approximation the per-round complexity is K^2 for arm selection and d^3 for the inversion of A. The per-round complexity of the convex relaxation is roughly constant for arm selection and again d^3 for the inversion of A, while the cost of solving the convex problem depends on the specific solver (eg, gradient descent). In this sense, the computational complexity of the proposed methods is not a major concern because it is similar to standard LB algorithms (eg, LinUCB has K complexity for arm selection and d^3 for matrix inversion). The approximation error of the implementations is discussed in Lemmas 5-8 and is reported as a (1+beta) factor in the analysis of the theorems.

5C Comparison to MAB algorithms and "choice of the setting"
When moving from MAB to LB the gain in complexity comes from moving from K to d. In our experiment, although the setting is almost a MAB, there is already a huge gap in the performance of XY-Adaptive and that of algorithms for best-arm identification in MAB (eg, Successive-Elimination(SE)). In fact, similar to XY-Adaptive, SE quickly discards all arms but x_1 and x_{d+1}. Then, while XY-Adaptive exploits the structure of the problem and pulls x_2 (thus improving the estimate of theta* along the dimension that better discriminates between x_1 and x_{d+1}), SE keeps pulling the remaining (near optimal) arms until it can confidently identify the best arm. This results in a much higher sample complexity for SE (order of 10^8 for d=2), even worse than static strategies. Also, the setting was chosen to create a big gap in the performance of XY-Oracle and the static strategies, which allowed to test how much XY-Adaptive is effective in moving from the initial static strategy to a more effective sampling strategy on "relevant" directions. This experiment provides a sanity check of the theory developed in the paper and is not intended to be a conclusive empirical evaluation of XY-Adaptive.